# Facial Image Encryption for Secure Face Recognition System

Eimad Abusham [1],*, Basil Ibrahim [1], Kashif Zia [2] and Muhammad Rehman [1]

1 Faculty of Computing and Information Technology, Sohar University, Sohar 311, Oman
2 School of Health & Wellbeing, University of Glasgow, Glasgow G20 7SB, UK
* Correspondence: eabusham@su.edu.om

**Abstract:** A biometric authentication system is more convenient and secure than graphical or textual passwords when accessing information systems. Unfortunately, biometric authentication systems have the disadvantage of being susceptible to spoofing attacks. Authentication schemes based on biometrics, including face recognition, are susceptible to spoofing. This paper proposes an image encryption scheme to counter spoofing attacks by integrating it into the pipeline of Linear Discriminant Analysis (LDA) based face recognition. The encryption scheme uses XOR pixels substitution and cellular automata for scrambling. A single key is used to encrypt the training and testing datasets in LDA face recognition system. For added security, the encryption step requires input images of faces to be encrypted with the correct key before the system can recognize the images. An LDA face recognition scheme based on random forest classifiers has achieved 96.25% accuracy on ORL dataset in classifying encrypted test face images. In a test where original test face images were not encrypted with keys used for encrypted feature databases, the system achieved 8.75% accuracy only showing it is capable of resisting spoofing attacks.

**Keywords:** face recognition; LDA; image encryption; cellular automata





## 1. Introduction

A Face Recognition algorithm uses machine learning techniques to detect and identify human faces by analyzing visual patterns in visual data [1]. One of the key advantages of facial recognition systems is that they allow users to be passively authenticated [2] that is, they allow users to prove their authenticity simply by being in the room without having to interact with the system at all. Video surveillance, access control, forensics, and social media are some of the many applications in which facial recognition systems are used.

As explained by [3], facial recognition systems have six stages. Preprocessing is the first stage. An area of interest is aligned when faces are detected in the visual input. Using preprocessed input, face features are extracted at the second stage. During the final stage, extracted face features are compared with features in the database for matching results. An identification is based on facial features stored in memory, or a verification is based on matching results.

The face recognition authentication system has a number of advantages and limitations, as all biometric subsystems. It is more secure to use biometric authentication than to use conventional password authentication [4]. As biometric traits cannot be forged, registration is required, preventing false authentication [5], and each person's data is unique [6]. An important drawback of biometric authentication systems is that they are susceptible to spoofing attacks, as well as attacks on deep learning and machine learning systems. In spoofing attacks, attackers present false biometric information in order to gain credibility [7]. Synthesis is considered a spoofing attack by [8]. According to [9,10], reply attacks can also be described as spoofing attacks. Machine learning and deep learning models can be attacked in a variety of ways, including adversarial attacks [4] and poisoning attacks [11]. In this paper a method for preventing spoofing attacks on face recognition system is proposed by integrating a model for image encryption on recognition process. Image

encryption model is used to encrypt preprocessed face images that are used to train and test a face recognition model based on Linear Discriminant Analysis (LDA) algorithm. Extracted features enrolled on features database are encrypted, meaning that in order to gain authenticity face image needs to be encrypted with correct image encryption key in order for classifier to correctly identify or verify submitted input. With that effectiveness of spoofing attacks are minimized; other than illegitimately obtaining copies of face images of authenticated individuals and submit it to system, an attacker needs to encrypt face image with correct key as well.

In order for image encryption models to provide this extra layer of security, they must offer high encryption performance and be resistant to brute force attacks. The image encryption model used in the recognition process is based on the XOR operation and a special type of Cellular Automata called the Outer Totalistic Cellular Automata (OTCA). XOR is applied to pixels bits for pixel substitution, while CA is used for image scrambling. Pixel substitution involves changing the original values of pixels using mathematical operations, and then applying the reverse operation to recover the original values [12]. By shuffling the original pixel locations on an image, image scrambling breaks the high correlation between pixels that were originally adjacent [13]. In order to generate complex structures from simple structures, CA can utilize simple structures [14], making it an excellent choice for image scrambling applications [15]. Most of studies on facial recognition systems focus on increasing recognition accuracy of model on various datasets however, not enough work is presented in addressing weaknesses of such systems including spoofing issues. This paper proposes a solution to spoofing issue on facial recognition through addition of an image encryption step in recognition process. The main goal is to build a facial recognition system that correctly classifies encrypted facial images for each subject in a selected faces dataset. The face recognition model is based on LDA and is trained using a training set formed of encrypted subjects' face images. Testing the model is conducted using two testing sets the first of which is formed of the remaining encrypted subjects' face images whereas the other will consist of the same remaining images however, in this case the images used are the original or decrypted images. Encryption of face images is implemented with new method based on XOR pixels substitution and scrambling based on CA. The contributions of this paper are as follows:

- The encryption of images consists of two main stages. Pixel substitution is the first stage of the process, during which each pixel value is substituted by a new value generated by performing an XOR operation on each pixel bit. A second stage involves shuffled pixel positions into new positions using CA during the pixels scrambling stage.
- A Linear Discriminant Analysis (LDA) is used to extract features from encrypted images and a Random Forest classifier is used to classify them. LDA reduces the dimensionality of the feature space to maximize the separation between classes by transforming the feature space. Consequently, more discriminative features are generated, improving the performance of the classifier. Random Forest is a classification method that uses multiple decision trees to classify data. The method has a high degree of accuracy and is robust to overfitting. Combining LDA and Random Forest classifiers makes for a powerful face recognition algorithm.
- The use of encrypted face images to train the model causes the model to recognize test images encrypted with the same encryption key only with high accuracy. This drastically reduces the effectiveness of spoofing attacks as an attacker would need to encrypt images with the same key in addition to obtaining an artificial copy of an authenticated individual face image.

The rest of this paper is organized as follows. On second section related work to image scrambling and face recognition with LDA is presented; third section explains implemented methodology; fourth section demonstrates results; last section concludes the study.

## 2. Related Work

Using universal rules, CA cells can change their state every discrete time step in response to their neighbors. According to [16], CA-based image encryption works directly on pixels to encrypt images. Aside from its ease of use, CA image encryption provides high security, parallel computing capabilities, and high performance [16]. In addition to image encryption, CA is capable of encrypting other types of information as well. Using a reversible CA based block cipher algorithm [17], proposed an algorithm that could handle CPUs with different core counts and supported scalability beyond 128 bits. The security framework offered by [18] is made up of three stages. The first is entity authentication with a zero-knowledge protocol, while the second and third stages are encryption and decryption with CAs. Several CA-based image scrambling techniques have been developed for use in image encryption [15,19–24]. Those studies found CA scrambling to be effective against a variety of attacks, breaking high rates of correspondence between adjacent pixels. It has been found that a large number of CA-based image scrambling techniques have been developed [14,18–23]. In those studies, CA scrambling performed well against different types of attacks, breaking high correlation between adjacent pixels. The two-dimensional CA was used by [15] for image scrambling. A number of different configurations, such as evolved generations, neighborhood configurations, boundary conditions, and rules with lambda values near critical values, were examined for their effect on *GDD* metric scrambling performance. By using all lattices evolved from the initial lattice, the method scrambles the image. An empty lattice is created, then on that lattice pixel locations that correspond to live cells on first scrambling matrix take the pixel's values of original subject image starting from top-leftmost cell then proceeding in row major order. For the remaining scrambling matrices, pixels at locations that have already been filled are skipped during the process. Pixels are copied from the original image to dead cells in row major order on the scrambling matrix. Based on the obtained results, higher generation scrambling results had better Gray Difference Degree (*GDD*). Additionally, there was a higher *GDD* achieved when Moore's configuration with periodic boundary conditions was used. Among the rules tested, lambda values ranged from 0.20703 to 0.41406. The highest *GDD* value on tested images was achieved by Rule 224—Game of Life.

It was investigated by [20] whether other 2D-OTCA rules could be applied to scramble images besides Game of Life. Instead of using Moore's rule, the authors use von Neumann neighborhood configuration. Different OCTA rules are experimented on and *GDD* is used for evaluating scrambling performance. Boundary conditions and generation selection are also taken into consideration. An initially generated lattice is evolved k times based on a randomly generated initial lattice. An image of the subject is scrambled using the final evolved lattice. On an empty lattice, pixel values of the original subject image are taken from the top-leftmost cell and proceed in row major order until every cell on the lattice corresponds to an alive cell on the scrambling matrix. By copying pixels from the original image to dead cells on the scrambling matrix, each pixel will correspond to a cell on the matrix in row major order. According to rule 171, this technique showed the highest *GDD* results of all the proposed techniques. As compared to other techniques, this technique took significantly less time to compute. Gray image encryption algorithm developed by [23] uses 2D CA to scramble the image at the bit level. Binary images representing bits are generated from an image by converting it into 8 binary images. An initial configuration lattice of 8 binary images is created and evolved using a B3/S1234 CA rule. As a result, 8 binary images of the original image are scrambled independently using evolved 8 binary lattices in the same way, thereby changing their positions and values simultaneously.

A modification to the 2D CA image scrambling technique proposed by [15] was implemented by [21], which resulted in better *GDD* scrambling. In the same way, all evolved lattices are used to scramble data. Using row major order, pixels on the original image that correspond to live cells on the scrambling matrix are copied to an empty lattice. On the remaining locations on row major order, the remaining pixels are copied as well. For the remaining scrambling matrices, repeat the same procedure. A scrambling lattice is

evolved according to Game of Life 224 rules. Based on the results, the best *GDD* is achieved when periodic boundary conditions, Moore's neighborhood, and more generations are included up to eight. Based on periodic boundary conditions, Moore's neighborhood, and Moore's neighbors, the highest *GDD* for eight generations was 0.9551.

In [19], CA was proposed for scrambling and watermarking images. Chaos can be detected in fractal CA rules by analyzing fractal box dimensions. After creating an initial lattice, a lattice is evolved based on a selected CA to scramble the image. This process scrambles the image as an initial step in watermarking. Furthermore, watermarked images produced using this scheme are less susceptible to noise attacks, cropping, and JPEG compression.

The author of [22] proposed using OTCA for encrypting images at the bit level. In rule 534 and rule 816, the bit values and locations of the original images are simultaneously changed with high computational efficiency at the bit level. An analysis of histograms and entropies indicates that the encryption method is robust. Furthermore, the key space is highly sensitive in addition to being large. It was found that each test image had an *NPCR* of nearly 100%, an Entropy of over 7.2, and a correlation almost equal to zero in each direction. Based on histogram analysis, encrypted images cannot be distinguished from their originals.

A method for scrambling images that uses ECA was proposed by [24]. ECA rules were used to test scrambling performance in classes 3 and 4. The scrambling method converts original images into 1D vectors. After k generations of evolution, a random 1D lattice is scrambled. Pixels are copied from the original image onto the empty 1D lattice and positioned where the live cells are on the scrambling lattice corresponding to pixels in the original image. Similarly, scrambling matrices with pixels already filled will skip matrices with unfilled pixels. As pixels are copied onto dead cells in the scrambling matrix, they correspond to the pixels that are still in the original image. An output 2D matrix is generated after converting a 1D vector to a 2D matrix. Using ECA for scrambling did not result in any difference in performance between *GDD* and 2D CA, and in some cases, performance was even better. A high *GDD* was obtained with Rule 22 when boundary conditions were combined with ten generation numbers. In class 3 rules tested (22, 30, 126, 150, 182), the GDDs were higher than in class 4 rules (rule 110).

As real-time processors become more common, research on automatic recognition of faces has become quite active, aiming to facilitate commercial applications by taking advantage of the human ability to recognize faces as special objects. The analysis of human facial images has been the subject of numerous studies. There are several ways in which facial features can be used to discriminate between people based on their gender, race, and age. In studies that used subjective psychovisual experiments, these features have been analyzed for their significance. Linear discriminant analysis (LDA) can be used to recognize faces by maximizing within-class scatter and minimizing between-class scatter through the combination of within- and between-class scatter. With LDA, different features of the face are objectively evaluated for their significance in identifying the human face. Using LDA for recognition can also yield a few features. LDA overcomes the limitations of Principle Component Analysis by using a linear discriminant criterion. By using this criterion, the projected samples' between-class scatter matrix is compared with their within-class scatter matrix in order to maximize that ratio. A linear discriminant used to classify images results in the separation of images into different categories.

A variety of methods for analyzing the features of the face are described in the literature based on local linear discriminants. Through the use of nonparametric discriminant analysis (NDA) and multiclassifier integration, Ref [25] developed a new framework for face recognition. The principal space and null space of the intra-class scatter matrix are being used to improve two multi-class NDA algorithms (NSA and NFA). The NFA uses classification boundary information more effectively than the NSA. Ref [26] also proposes enhancing order-based coding capabilities to increase intrinsic structure detection in facial images in addition to enhancing local textures. By selecting the most discriminatory

subspace, multimodal features can be automatically merged. In order to produce robust similarity measurements and discriminant analyses, adaptive interaction functions are used to suppress outliers in each dimension. In order to address the classification issue raised by a compact feature representation, the sparsity-driven regression model is modified. "Exponential LDE" (ELDE) is a new discriminant technique introduced by [27]. ELDE can be viewed as a compromise between the two-dimensional extension of LDE and the local discriminant embedding (LDE). Using the proposed framework, the SSS problem is overcome by eliminating the null space associated with locality-preserving scatter matrices. Distance diffusion mapping transforms original data into a new space and then applies LDE to the new space, similar to kernel-based nonlinear mapping. Increased margins between samples of different classes improve classification accuracy. The [28] method uses the local geometry structure of the data while applying a globally discriminatory structure from linear discriminant analysis, which maximizes between-class scatter while minimising within-class scatter. In kernel feature spaces, nonlinear features can also be produced through the optimization of an objective function.

A new ensemble approach for discovering discriminative patterns has been developed by [29]: the many-kernel random discriminant analysis (MK-RDA). In the proposed ensemble method, the authors incorporate a salience-aware strategy whereby random features are chosen on the semantic components of the scrambled domain using salience modeling. By optimizing binary template discriminability, Ref [30] proposes a new binarization scheme, using a novel binary discriminant analysis, a real-valued template can be transformed into a binary template. Because binary space is hard to differentiate, direct optimization is challenging. In order to solve this problem, a binary template discriminability function was developed using the perceptron.

## 3. Methodology

This section details procedures involved in integration of image encryption technique into pipeline of LDA based face recognition system. Firstly, phases related to processing images to obtain their encrypted versions are elaborated, following that LDA algorithm steps are explained, and modified face recognition scheme pipeline is expounded with LDA and encrypted features database.

### 3.1. Image Encryption Scheme

Image encryption technique is implemented with two phases. In the first phase pixels' values are substituted using XOR operation. On second phase pixels' positions are shuffled based on 2D lattice combined from different generations of lattices evolved from same initial lattice using CA. Initial lattice is randomly generated grid of cells confined within defined space. Using appropriate CA rules initial lattice is evolved yielding new lattices with each generation.

#### 3.1.1. XOR Pixels Substitution

Digital images are composed of pixels usually with different intensities that are represented in numerical range depending on image' bit depth. For example, if an image bit depth is 8 bits then pixels are represented with 8 bits then their numerical values range from 0 to 255. Note that longer bit depth as in case for RGB images describe bit depth for three channels with each channel can be uniquely represented with third of bit depth length. Then pixels can be represented as binary strings. Pixels substitution on proposed method is obtained by applying logical XOR operation between adjacent binary bits. If binary string length is $n$ then applying operation as explained would generate $n-1$ string, however the left most bit is kept the same as original binary string therefore generated binary string will have length of $n$. The conversion of any binary string with XOR is demonstrated with algorithm below.

As such generally for any binary string $b$ with length $n$, the XOR representation $X$ for $b$ can be obtained as follows:

$$b = b_0 b_1 b_2 \dots b_{n-1} \tag{1}$$

$$X = b_0 (b_0 \oplus b_1)(b_1 \oplus b_2)(b_2 \oplus b_3) \dots (b_{n-2} \oplus b_{n-1}) \tag{2}$$

In Equation (1), a binary string $b$ consists of $n$ bits, and the position of each bit is indicated by a subscript. For instance, the first bit in binary string $b$ is $b_0$, the second bit is $b_1$, and so on. A binary string $b$ can be converted to an XOR representation $X$ using Equation (2). Since the first bit of $X$ is the same as $b$, which is shown in step 1 of Algorithm 1, the first bit of $X$ is $b_0$. A second bit is determined by XOR operations between $b_0$ and $b_1$, a third bit by XOR operations between $b_1$ and $b_2$, and so forth.

| **Algorithm 1:** Processing binary string with XOR to obtain new XORed string | |
|---|---|
| **Input:** $b$ [$n$ bits binary string] | |
| **Output:** $X$ [XOR on bits of $b$ ] | |
| 1 | Set $n = Length(b)$ [n value is number of bits in b] |
| 2 | Set $X = b[0]$         [leftmost bit of X is same as b] |
| 3 | Set $i = 0$ |
| 4 | $while(i < n - 1)\{X = X + b[i] \bigoplus b[i+1]\ i = i+1\}$ [XOR is applied between adjacent bits and appended to X] |
| 5 | Return $X$ |

As pixels values for any digital image are binary strings with length $n$, the same process can be applied to digital images converting an image to its XOR version. Whether the image consists of single channel (e.g., grayscale image) or multiple channels (e.g., RGB image) the process can be applied to each channel separately and channels can be recombined yielding XOR image. That is for any pixel at position $(i, j)$ in an image $I$, its value in XOR image $I'$ version of $I$ can be expressed by:

$$I'(i,j) = XOR(I(i,j)) \tag{3}$$

Given any X string its corresponding original binary string can be regenerated using XOR operation as well however it's applied with slight differences. Since the leftmost bit in $X$ is the same as in original $b$ then $b_0 = X_0$. The next bit $b_1$ is $b_0 \oplus X_1$, similarly $b_2$ is $b_1 \oplus X_2$ and the same process is repeated for $n-1$ times. The steps for converting XOR string to original binary string are shown below:

| **Algorithm 2:** Obtaining original binary string from its XORed version | |
|---|---|
| **Input:** $X$ [XOR string with length $n$] | |
| **Output:** $b$ [Original binary string] | |
| 1 | Set $n = Length(X)$ [n value is number of bits in X] |
| 2 | Set $b = X[0]$          [leftmost bit of b is same as X] |
| 3 | Set $i = 0$ |
| 4 | $while(i < n - 1)\{$ $b = b + b[i] \bigoplus X[i+1]$ $i = i+1\}$ [XOR is applied between adjacent bits and appended to b] |
| 5 | Return $b$ |

As such generally for any XOR string $X$ with length $n$, the original binary string $b$ corresponding to $b$ can be obtained as follows:

$$X = X_0 X_1 X_2 \dots X_{n-1} \tag{4}$$

$$b = X_0 \Rightarrow b_0 \Rightarrow b_0(b_0 \oplus X_1) \Rightarrow b_0b_1(b_1 \oplus X_2) \Rightarrow \dots \Rightarrow b_0b_1 \dots (b_{n-2} \oplus X_{n-1}) \quad (5)$$

Equation (4) represents XOR string $X$ that needs to be converted back to original binary string $b$. $X_0$ is the first bit of $X$, $X_1$ is the second bit of $X$ and so on until last $n$th bit which is $X_{n-1}$. Equation (5) shows how the binary string is obtained back in series of steps where the arrow indicates the next step. Initially binary string value contains $X_0$ which is the same as the original binary string as such the first bit becomes $b_0$. The next step shows how to obtain the second bit $b_1$ which is an XOR operation between $b_0$ and the second bit of $X$. This series of steps continue until the last bit is calculated.

Now that the conversion from XOR string to binary string is shown, the process can be applied to pixels of XOR image $I'$ to obtain original image $I$. That is for any pixel at $(i, j)$ in $I'$ the original pixel value can returned by:

$$I(i,j) = UnXOR\big(I'(i,j)\big) \quad (6)$$

3.1.2. Image Scrambling with CA

After altering pixels' values with XOR pixels substitution in first phase, the resulting image is scrambled based on 2D lattice combined from two other lattices evolved from same initial lattice using 2D-OTCA (Two Dimensional Cellular Automata) rules. OTCA are variation of CA rules introduced by [13] where a cell in a 2D lattice transit between predetermined set of states (e.g., alive or dead) based on the state of the cells in its neighborhood and the current state of that cell. OTCA rule can be reapplied to resulting lattice again to obtain new lattice and process can be repeated $k$ times to obtain $k$th generation lattice. The transition function for any pixel at coordinates $(i.j)$ in lattice $L$ in next iteration can be expressed with function $v$ as shown next:

$$I^{t+1}(i,j) = v\left(I^t(i,j), \sum_{i'j'} I^t(i',j')\right) \quad (7)$$

where $I^t(i', j')$ are cells in $I^t(i, j)$ neighborhood.

Neighborhood Configuration

As described in (7) transition function requires states of neighboring cells for determination of state of subject cell in subsequent generation of lattice. Most common neighborhood configurations are Von Neumann and Moore neighborhood schemes [31]. In Von Neumann Neighborhood scheme (denoted as $N_{vN}$) neighboring cells are adjacent cells in four cardinal directions (Figure 1). Given a radius $r$ neighborhood range can be extended, with that a cell at coordinate $(i', j')$ neighbors a cell at coordinate $(i, j)$ in Von Neumann configuration if it satisfies the following rule:

$$(i',j') \in N_{vN}(i,j,r) \ if \ |i'-i| + |j'-j| \leq r \quad (8)$$

Moore's Neighborhood scheme (denoted as $N_M$) differs from $N_{vN}$, it includes diagonally adjacent cells to subject cell as part of the neighborhood. Similarly given a radius $r$ neighborhood range can be extended, with that a cell at coordinate $(i', j')$ neighbors a cell at coordinate $(i, j)$ in Moore's configuration if it satisfies the following rule:

$$(i',j') \in N_M(i,j,r) \ if \ |i'-i| \leq r \ and \ |j'-j| \leq r \quad (9)$$

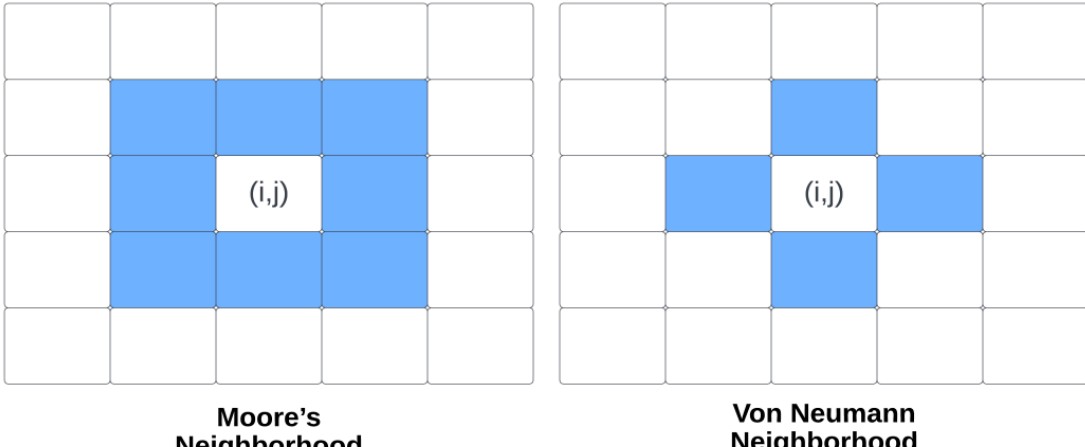

**Figure 1.** Illustration of Moore's and Von Neumann neighborhood schemes at radius 1.

Boundary Conditions

Cells within lattices are confined within 2D rectangle space. Determining neighboring cells for border cells is resolved with two conditions; either Closed Boundary Condition (CBC) or Periodic Boundary Condition (PBC) [32]. In CBC missing cells from neighborhood are given naught state [33]. As for PBC cells at borders are adjacent to each other; that is for instance cells at top are adjacent to bottom cells, leftmost cells are adjacent to rightmost cells and corner cells are adjacent to each other.

Conway's Game of Life

Conway's Game of Life (CGL) credited to John Conway, is most universally known automata [34]. CGL is an OTCA rule which determines cell's next state based on current cell's state and neighboring cells in $N_M$ configuration at $r = 1$. As stated by [35] in CGL or any similar variation neighboring cells are cells that are directly touching subject cell. As such neighboring cells in CGL are cells that confirms to $N_M$ configuration at $r = 1$ of subject cell.

In CGL cells can transition between states—alive (1) or dead (0)—in subsequent evolutions. Transition of cell state between alive or dead is determined from the following rules:

- For a cell at coordinates $(i, j)$ such that $I^t(i, j) = 0$ if $\sum_{i', j'} I^t(i', j') = 3$ for $(i', j') \in N_M(i, j, r)$ then $I^{t+1}(i, j) = 1$ otherwise $I^{t+1}(i, j) = 0$
- For a cell at coordinates $(i, j)$ such that $I^t(i, j) = 1$ if $2 \leq \sum_{i', j'} I^t(i', j') \leq 3$ for $(i', j') \in N_M(i, j, r)$ then $I^{t+1}(i, j) = 1$ otherwise $I^{t+1}(i, j) = 0$

Figure 2 shows an example of lattice evolved with CGL rules. Alive cells are illustrated in white and dead cells are in black. The boundary condition in the illustration is set as PBC and neighborhood scheme is Moore's scheme.

Scrambling Algorithm

Image scrambling process starts by generating lattices required for shuffling image pixels into new positions. Initially a lattice with random cells' states and dimensions equal to original image dimensions is generated. Next the lattice is evolved with (7) such that neighborhood is $N_M$, boundary condition is PBC and CGL defines new state of subject cell on next evolution. That is transition function becomes:

$$I^{t+1}(i, j) = v_{CGL}\left(I^t(i, j), \sum_{i'j'} I^t(i', j') \mid I^t(i', j') \in N_M(i, j, r)\right)_{PBC} \tag{10}$$

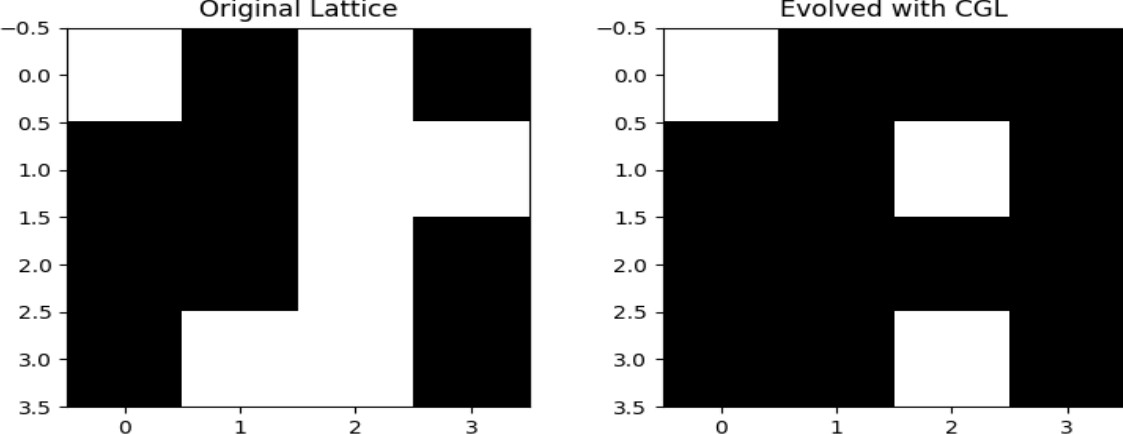

**Figure 2.** An example of a lattice evolved with CGL rule (Alive cells are white and black cells are dead). PBC and $N_M$ are applied.

Lattice is evolved to $k$th generation, then $k$th generation lattice is combined with one of previous generation which is named $n$th generation. This new combined lattice is scrambling lattice that is used to scramble image pixels. The steps for scrambling algorithm are elaborated in Algorithm 3 and an abstractive illustration of the proposed scheme in on Figure 3.

| **Algorithm 3**: Encryption algorithm | |
|---|---|
| Input: *I* [Original image] <br> Output: *E* [Image after scrambling] | |
| 1 | Convert pixels in *I* to its corresponding XOR string equivalent in *I'*. That is value of pixel at coordinates $(i, j)$ in *I'* is obtained by $I'(i, j) = XOR(I(i, j))$ |
| 2 | Generate random lattice $L_0$ with exactly the same width and height as *I*. Values of lattice pixels can either be 1 (alive) or 0 (dead). |
| 3 | Apply CGL OTCA transition function $v_{CGL}$ on on$L_0$ with $N_M$ and PBC for k generations yielding $L_k$. |
| 4 | Combine $L_k$ and $L_n$ $(0 < n < k)$ on an initially empty lattice *R* such that $R(i, even(j)) = L_k(i, j)$ and $R(i, odd(j)) = L_n(i, j)$. |
| 5 | Transform *I'* into a stack such that elements in stack from top to bottom are values of pixels in *I'* in row major order. |
| 6 | Scramble *I'* such that: search *R* in column major order and if $R(i, j) = 1$ pop an element from top of $stack(I')$ into initially empty scrambled image *E* at same coordinates $(i, j)$. After search is complete search R again in row major order this time and if $R(i, j) = 0$ pop an element from top of $stack(I')$ into scrambled image *E* at same coordinates $(i, j)$. |

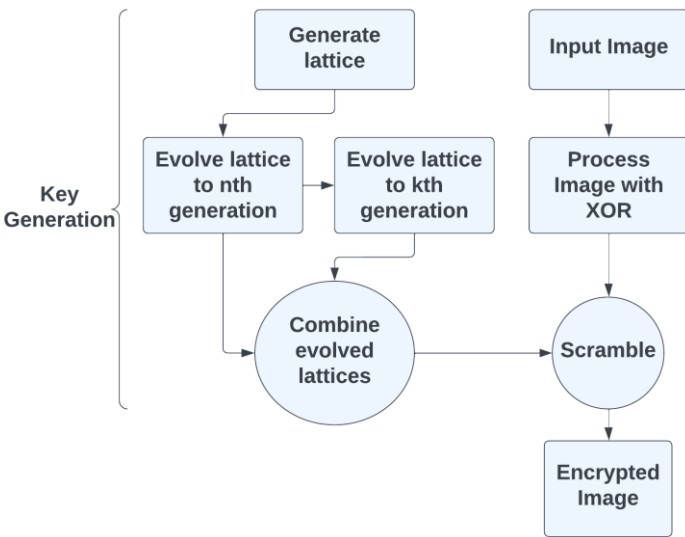

**Figure 3.** An illustration of main steps of proposed image encryption scheme.

Now that Algorithm 3 is elaborated encryption key used to obtain encrypted version of original image can be expressed. The encryption key consists of five main elements:

- Randomly generated lattice $L_0$. This lattice is created with same width and height of image that require encryption. Pixels in $L_0$ can assume two states only either alive (white) or dead (black).
- The CA rule used to evolve $L_0$. In proposed image encryption scheme the CA rule used is CGL rule however, different rules can be utilized to evolve $L_0$ to desired number of generations. Neighborhood structure can be changed as well for example using $N_{vN}$ or extending radius of neighborhood. In this work the neighborhood structure is $N_M$ with radius 1 due to requirements of CGL.
- Boundary condition used to handle cells at borders for $L_0$ or any subsequent evolved lattices. In proposed technique PBC was selected however, CBC can be selected as well.
- The number of generations $n$ selected to evolve $L_0$ to obtain $L_n$. The value of $n$ is bounded to be more than 0 and less than another value $k$ i.e., that $0 < n < k$.
- The number of generations $k$ selected to evolve $L_0$ to obtain $L_k$. Now that since $0 < n < k$ then $L_k$ is rather obtained by evolving $L_n$ for $(k - n)$ times.

In order to show that encryption keys are robust, their ability to resist brute force attacks must be analyzed. When using brute force attack to decrypt an encrypted image, it is necessary to recreate the randomly generated lattice $L_0$ to precisely match the initial cell states, since slight differences in initial states would influence neighboring cells' states in subsequent evolutions $L_1$, and those neighboring cells will influence the evolution of their own neighboring cells, and the effect continues with each evolution. Also recreation of $L_0$ to exact initial cells states becomes more difficult with increasing size of $L_0$ as more initial cells states are accounted for. Initial states of cells are one of two—either alive or dead—and number of cells is determined from size of lattice $\left(width_{L_0} \times height_{L_0}\right)$ then brute forcing $L_0$ recreation could require $2^{size(L_0)}$ recreation tries. The number of generations $n$ and $k$ are selected to evolve $L_0$ to $L_n$ and $L_k$ then the resulting evolutions are combined into scrambling lattice $R$. To break encryption key with brute force attack given that $L_0$ is available, $L_0$ is evolved $k$ times then its combined with each $L_n$ where $0 < n < k$. The process continue until the correct value of $k$ is encountered and right combination with $L_n$ is found to produce $R$. Given that repetitions of $L_n$ and $L_k$ combinations are eliminated during the search for $k$ then number of actual unique combinations tried equals size of unique pairs of $n$ and $k$ set at correct value of $k$. The size of such set (denoted as $u$) is $k(k-1)/2$. Then

for an attacker to break encryption key with brute force attack the descrambling algorithm needs to run up to $u \times 2^{size(L_0)}$ times.

The encryption algorithm utilizes CGL with $N_M$ at radius 1 and PBC for evolving $L_0$. However, those configurations can be changed and their inclusion in main components of encryption key indicates the possibility. There are many rules for OTCA and rules classified as Class 3 and 4 generate different and complex patterns in each evolution [31]. Also neighborhood structure can be extended or changed and with different structures comes different patterns in next evolutions. Boundary condition selection as PBC or CBC affect generated patterns in subsequent generations of initial lattice $L_0$ as well. If such configurations are allowed to be changed then breaking encryption key with brute force attack would require significantly more than $u \times 2^{size(L_0)}$ times. Since the encryption algorithm is demonstrated with CGL, $N_M$ at radius 1 and PBC then in this regard the key space is $u \times 2^{size(L_0)}$.

With five elements composing the encryption key are explained, then generation of encrypted image $E$ can be expressed as function $e$ in Equation (11) that takes XOR image $I'$ and key configurations required for encryption as its parameters.

$$E = e(I', L_0, v_{CGL}, PBC, k, n) \tag{11}$$

Descrambling Process

Descrambling requires regeneration of scrambling lattice $R$. Given scrambling key, $R$ is recreated which is then used to reorder shuffled pixel positions of $R$ back to $I'$. Then XORed pixels values can be 'UnXORed' to retrieve original pixels' values. Steps for descrambling $E$ are shown in Algorithm 4:

---

**Algorithm 4**: Descrambling algorithm

---

Input: $E$ Scrambled image
Output: $I'$

| | |
|---|---|
| 1 | Generate $R$ from provided keys where $R = f(L_k, L_n)$. |
| 2 | Search $R$ in column major order if $R(i,j) = 1$ then $E(i,j)$ is added to $stack(E)$. |
| 3 | Search $R$ in row major order if $R(i,j) = 0$ then $E(i,j)$ is added to $stack(E)$. |
| 4 | Reverse $stack(E)$ then pop elements from stack in an initially empty lattice generating $I''$. |

---

Descrambling can also be expressed as a function d in Equation (12), where E is the parameter, and encryption key configurations are the input parameters. To retrieve original pixel values, the UnXOR function detailed in Algorithm 2 is used to process the output of function $d$.

$$I' = d(E, L_0, v_{CGL}, PBC, k, n) \tag{12}$$

*3.2. LDA Algorithm*

Linear Discriminant Analysis (LDA) proposed by R. Fisher is one of fundamental data analysis methods [36]. LDA is deployed for features extraction and dimensionality reduction [37] in applications such as face recognition [38], retrieval of images [39], and microarray classification of data [40].

LDA is largely adopted for problems than involves dimensionality reduction [41]. LDA is a suitable machine learning algorithm for reducing dimensionality in classification problems that involve more than two classes. LDA is efficient for multi-class classification as it can be used for data pre-computation to reduce the number of features, thereby reducing computational costs. LDA is preferred over other approaches. In LDA, the variance of projections within a class has the lowest variance, and the variance of projections

outside the class has the highest variance. In PCA, all data are treated equally, resulting in very representative projected data, but it may sometimes obfuscate discrimination between classes.

Here LDA is used for features extraction of encrypted face images. Extracted features are compared with features database to classify input encrypted face images into appropriate classes. Input face image is encrypted with the same key used for encryption of features database. The steps of LDA implementation in context of proposed pipeline are shown below:

1. Preprocessing faces image dataset with face detection and alignment. Encrypt dataset and scale it then split dataset into training set and testing set.
2. Convert 2D encrypted face images training dataset into 1D vectors $\{F_1^1, F_2^1, \ldots, F_n^c, \ldots\}$ where $n$ is $n$th image of class $c$. Each face image 1D vector is image's *width × height* long.
3. Find mean vector for each class $k$ by $u_k = \frac{F_1^c + F_2^c + \cdots + F_n^c}{n}$. Then the set of all mean vectors is $\{u_1, u_2, \ldots, u_k, \ldots\}$. The length of this set is equal to number of classes.
4. Calculate overall mean vector for all classes $u = \frac{u_1 + u_2 + \cdots + u_n}{n}$.
5. Find between class scatter matrix $S_b = \sum_{k=1}^{c} n_k (u_k - u)(u_k - u)^T$ where $n_k$ is number of samples per class and $c$ is number of classes.
6. Determine within class scatter matrix $S_w = \sum_{k=1}^{c} \sum_{F_i^c} (F_i^c - u_k)(F_i^c - u_k)^T$.
7. Obtain linear discriminants by solving for Eigen values and vectors for matrix $S_w^{-1} S_b$ with single value decomposition.

### 3.3. Encrypted Face Recognition with LDA

After presentation of image encryption process and LDA algorithm a complete pipeline for the proposed encrypted face recognition can be illustrated as shown in Figure 4. The process starts by preprocessing face images into correct alignment and size. However, this step is not mandatory for datasets with prepared alignment and face detection such as ORL dataset. Next entire face images database is encrypted with single key and divided into training set and testing set. Training set features are extracted with LDA and stored in features database then they are used for classification of test set of encrypted face images with random forest classifier.

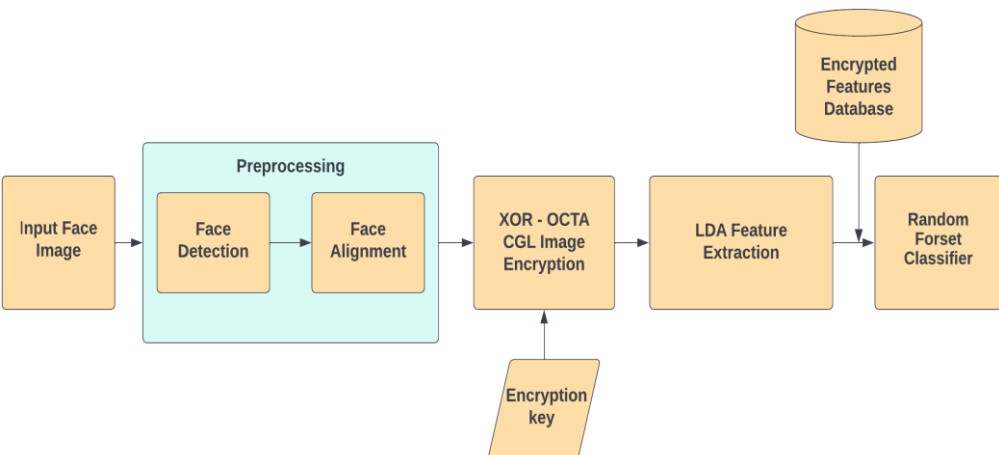

**Figure 4.** LDA Face Recognition scheme with Encrypted Face Images.

Random forest classifier, introduced in 2001 by Leo Breiman [42] consists of independent set of decision trees that collectively classify input data based on majority votes [43]. In random forest classification random samples are selected from training set and an independent decision tree is generated for each sample. Each decision tree outputs its prediction and based on majority votes for predications a classification output is given.

## 4. Results and Discussion

Experimentations on XOR-OTCA CGL image encryption method and recognition of encrypted face images with LDA were conducted on standard laptop with 8 GB RAM, Intel(R) Core(TM) i5-3230M CPU @ 2.60 GHz and Microsoft Windows 10 Home 64 bits using Python3. Firstly, the robustness of image encryption scheme is measured using differential analysis with *NPCR* (Number of Pixels Change Rate) and statistical analysis with histogram, correlation and key sensitivity. *GDD* (Gray Difference Degree) is also measured for comparison with other proposed methods in literature.

After demonstration of robustness of image encryption scheme, ORL face dataset is utilized for measurement of LDA with random forest classifier accuracy in classification of encrypted face image. Since ORL dataset is preprocessed with face detection and alignment, the entire dataset is directly encrypted with same key and inserted into pipeline of encrypted face recognition with LDA post preprocessing stage.

### 4.1. Analysis of Image Encryption Scheme

#### 4.1.1. Illustration of Gray Code OTCA CGL Encryption and Decryption on Sample Image

In this subsection, a sample grayscale image is encrypted using arbitrary selected key configurations to demonstrate the processes applied by the proposed scheme to generate the encrypted image. Figures 5 and 6 show encryption and decryption process implemented on sample image. As shown in Figure 5 sample image is firstly processed with XOR pixels substitution converting it to its XORed version. Initial scrambling lattice with dimension identical to sample image is generated with random cells states and evolved with CGL rules to nth and kth generation. Then R is generated as described in encryption algorithm and used for scrambling XORed sample image. In decryption process illustrated on Figure 6 the encryption key is used for descrambling encrypted image yielding XORed sample image and finally XORed sample image is UnXORed to obtain original sample image.

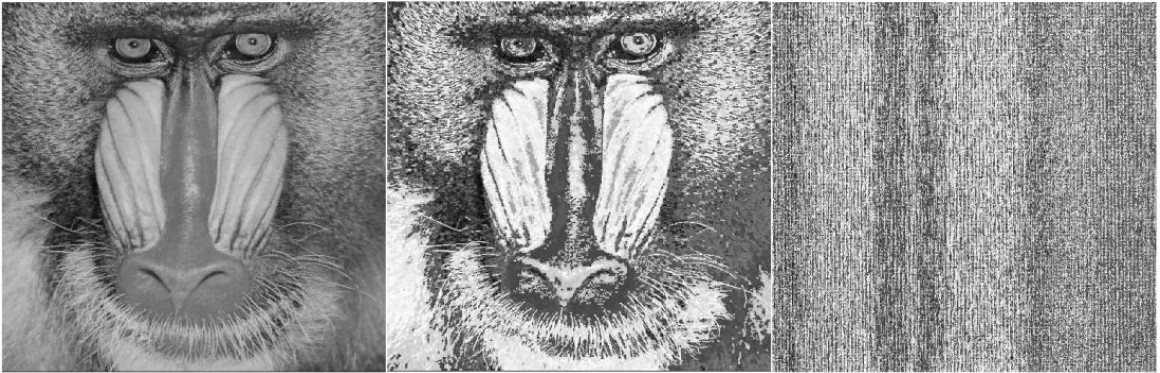

**Figure 5.** Encryption of Mandrill image. From left original grayscale image, XORed Image, and scrambled image with configurations $R = e(I', L_0, v_{CGL}, PBC, 12, 7)$.

#### 4.1.2. Differential Analysis

##### Number of Pixels Change Rate

Number of Pixels Change Rate (*NPCR*) measures image encryption technique resistance to differential attacks [32]. The closer percentage of NPRC of encrypted images to 100% the more robust the encryption technique is. Given original image and encrypted version of image *NPCR* can be determined as follows:

$$NPCR(I, E) = \frac{\sum_{i=0}^{width(I)-1} \sum_{j=0}^{height(I)-1} x(i,j)}{resolution(I)} \; if \; I(i,j) = E(i,j) \; \rightarrow x(i,j) = 0 \; if \; I(i,j) \neq E(i,j) \; \rightarrow x(i,j) = 1 \qquad (13)$$

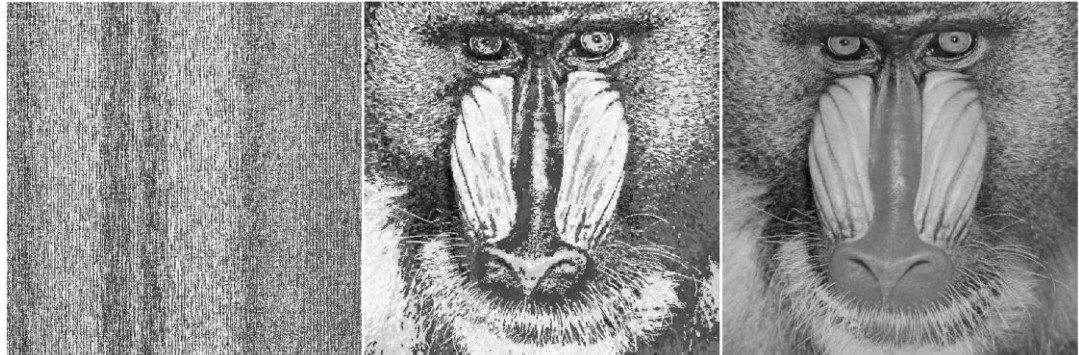

**Figure 6.** Decryption of Mandrill image. From left scrambled image, descrambled XORed image with configurations $I' = d(E, L_0, v_{CGL}, PBC, 12, 7)$, and original image (UnXORed image).

*NPCR* was computed for selected group images by comparing original images and their encrypted versions. The same key configurations were used for encrypting images; that is the same initial lattice $L_0$ was evolved for $n$th and $k$th generation with CGL and $R$ was created by combining $L_n$ and $L_k$. Table 1 shows that *NPCR* for encrypted group of images was more than 99% in all cases which shows the robustness of the technique.

**Table 1.** *NPCR* and *UACI* between Original Test Images and their Scrambled Versions. For any encrypted image $E_{im} = e(I'_{im}, L_0, v_{CGL}, PBC, 10, 8)$.

| Test Image | NPCR | UACI |
| --- | --- | --- |
| Lena | 99.597% | 28.07% |
| Cameraman | 99.467% | 37.075% |
| Barbara | 99.908% | 32.736% |
| Mandrill | 99.688% | 29.284% |
| Peppers | 99.489% | 31.934% |
| Airplane | 99.596% | 19.904% |
| Gold hill | 99.426% | 24.525% |

To demonstrate the proposed method effectiveness in producing large *NPCR* values, the experiment was repeated on entire Yale dataset. Yale dataset consists of 15 subjects with 11 images per subject, totaling 165 images. Obtained values for *NPCR* were all above 99%, Table 2 shows some of *NPCR* results obtained for different subject's images. Appendix A, Table A1, contains the full results.

**Table 2.** Sample of *NPCR* and *UACI* between Original Test Images and their encrypted versions in Yale dataset. For any encrypted image $E_{im} = e(I'_{im}, L_0, v_{CGL}, PBC, 14, 10)$.

| Test Image | NPCR | UACI |
| --- | --- | --- |
| Subject 1–7 | 99.561% | 21.891% |
| Subject 3–4 | 99.344% | 36.389% |
| Subject 7–6 | 99.777% | 43.831% |
| Subject 2–8 | 99.752% | 39.081% |
| Subject 12–3 | 99.645% | 36.404% |
| Subject 9–10 | 99.808% | 41.877% |
| Subject 5–11 | 99.796% | 38.430% |

Universal Average Changing Intensity

Universal Average Change Intensity (*UACI*) is utilized here to find average intensity difference between original image and its encrypted version. Ideal value of *UACI* is 33% [44] and it's determined here with:

$$UACI = \frac{\sum_{i=0}^{width(I)-1} \sum_{j=0}^{height(I)-1} |I(i,j) - EI(i,j)|}{255 \times resolution(I)} \times 100 \qquad (14)$$

Similar to *NPCR*, *UACI* was computed for the same group of images using original images and same encrypted images used in *NPCR* test. Table 1 shows obtained *UACI* for encrypted group of images. *UACI* values for most images were around 33% whereas Airplane and Gold hill were notably lower. Again the *UACI* test was conducted for Yale faces dataset showing the results obtained in Table 2 and full results can be found on Table A1.

4.1.3. Statistical Analysis

Histogram

Histogram illustrates how pixels of different intensities are distributed in an image. Due to altering pixels' intensities in scrambled image, histogram of scrambled image is different than original image. This is reducing the possibility of identifying a specific encrypted image in set of encrypted images by matching histogram of original image to images stored in database, given that database of encrypted images become available to the attacker. Figure 7 shows histogram of Mandrill image and encrypted Mandrill image. More histograms can be found in Figure A1.

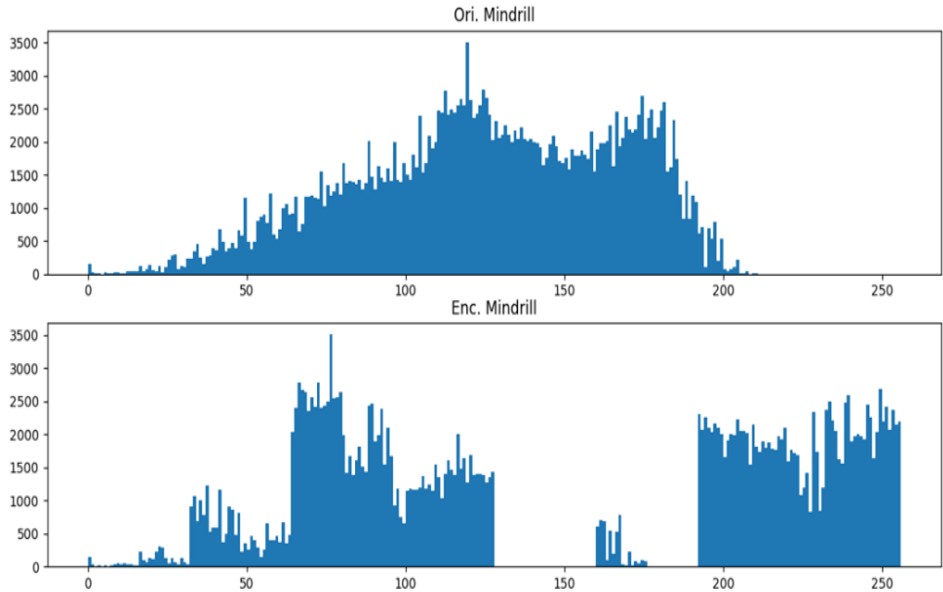

**Figure 7.** Histogram of Mandrill test image. On top Mandrill before scrambling and on bottom Mandrill after scrambling.

Correlation

Correlation indicates similarity between original image and its scrambled version. Correlation value varies between −1 to 1 inclusive. Having correlation index closer to 1 indicates that there is strong positive correlation between datasets and opposite (closer to −1) shows there is strong negative correlation. At 0 correlation, dissimilarity between images is highest as 0 indicates no correlation at all. As such the closer the correlation value to 0 is the more dissimilar the original image and its scrambled version are. Using

Karl Person's correlation formula correlation between several images and their scrambled versions is calculated.

$$r = \frac{\sum xy}{\sqrt{\sum x^2}\,\sqrt{\sum y^2}} \tag{15}$$

On proposed method same value of k and n were used to encrypt images for testing, however initial $L_0$ lattice was different for each tested image. That is $E_{im} = e\left(I'_{im}, L_0^{im}, v_{CGL}, PBC, 12, 6\right)$. Results on Table 3 show that there is approximately no correlation between test images and their scrambled versions for cases of entire image or subsections taken horizontally, vertically or diagonally.

**Table 3.** Correlation between Original Test Images and their Scrambled Versions.

| Test Image | Correlation | | | |
|---|---|---|---|---|
| | **Full** | **Horizontal** | **Vertical** | **Diagonal** |
| Lena | 0.034 | −0.041 | −0.017 | −0.031 |
| Cameraman | 0.108 | 0.032 | 0.127 | −0.083 |
| Barbara | 0.030 | 0.021 | 0.006 | 0.102 |
| Mandrill | −0.017 | −0.013 | −0.029 | 0.025 |
| Peppers | −0.013 | −0.027 | −0.104 | 0.022 |
| Airplane | 0.031 | 0.032 | 0.023 | 0.023 |
| Gold hill | 0.010 | −0.052 | 0.055 | 0.039 |

Key Space and Sensitivity Analysis

To test effectiveness of image encryption key space must be large enough to withstand brute force attacks [21]. For proposed algorithm the key is composed of initial $L_0$ lattice of size *width* × *height*, number of generations k and value n such that $0 < n < k$. Since *n* is selected randomly based on *k*, and pixels on initial $L_0$ lattice can assume one of two states (alive or dead) then key space is $u \times \left(2^{res(L_0)}\right)$, where $u = k(k-1)/2$ is the size of unique pairs of *k* and *n* set.

For instance if an image of size 256 pixels width and height requires encryption and value of k was selected to be 6 then key space in this case is $15 \times 2^{65536}$. This key space is exceptionally wide and with larger images and *k* values (which in turn increases the size of *u*) the key space increases exponentially. For small images greater value of *k* can be selected to widen key space.

To test sensitivity of key, encrypted Mandrill image on Figure 3 is decrypted using $L_k$ and $L_n$ only. Then keys with different values of *k* and *n* are tested to decrypt the image. Results on Figure 8 show that decrypting image is only possible with correct key. Given that $L_0$ is available decrypting image with $L_k$ or $L_n$ only yields no useful information and same can be concluded for different values of *k* and *n*.

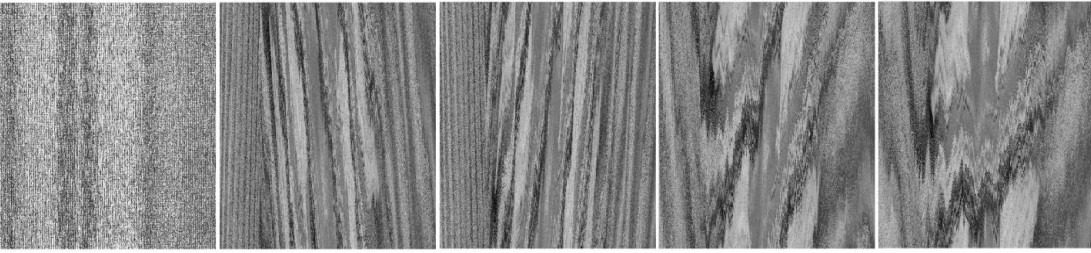

**Figure 8.** Decrypting Mandrill image with different values of *k* and *n*. From left encrypted image, decryption with $L_k$ only, decryption with $L_n$ only, decryption with *k* = 12 *n* = 6 and decryption with *k* = 12 *n* = 7.

Information Entropy

Information entropy measures the average information conveyed by image pixels [13]. For encrypted images in ideal cases pixels have uniform distribution with equal probability of occurrence [45]. Entropy is determined with

$$H(S) = -\sum_s ((P(s_i) \times \log_2 P(s_i)))$$

(16)

Since experimentation is carried out on grayscale images with 8 bits' depth (single channel) then maximum and ideal entropy value is 8. As such the greater the information entropy of ciphered image the more secure it is. Table 4 shows information entropy values for same encrypted images utilized in correlation test.

**Table 4.** *GDD* between Original Test Images and their Scrambled Versions and information entropy for encrypted images.

| Test Image | GDD | Information Entropy |
|---|---|---|
| Lena | 0.9650 | 6.8731 |
| Cameraman | 0.9793 | 6.0497 |
| Barbara | 0.8117 | 5.3528 |
| Mandrill | 0.8652 | 7.2789 |
| Peppers | 0.9686 | 7.5712 |
| Airplane | 0.9133 | 6.7024 |
| Gold hill | 0.9531 | 7.4778 |

Gray Difference Degree

*GDD* measures how well scrambled original image is after applying scrambling techniques. *GDD* metric was introduced by [19] and determining its value involves several steps. The first step is finding Gray difference *GD* for each pixel in both original image and scrambled/encrypted image. *GD* is found for all input image pixels except for pixels at edges. Also calculating *GD* requires neighboring pixels for subject cell in $N_{vN}$ neighborhood structure at radius 1. Determining *GD* is done by Equation (17).

$$GD(i,j) = \frac{1}{4} \sum_{i',j'} [P(i,j) - P(i',j')]^2$$

(17)

where $(i,j)$ is not a coordinates at edge of input image and $(i',j') \in N_{vN}(i,j,1)$ $if$ $|i'-i| + |j'-j| \leq 1$.

Next step is find Average *GD* for original image $Avg(GD(i,j))$ and scrambled/encrypted image $Avg_E(GD_E(i,j))$. Using the *GD* values obtained from Equation (17) enables calculating the value of Equations (18) and (19). Note that both equations are essentially the same however, the difference is on the context in which both equations values are found.

$$Avg(GD(i,j)) = \frac{\sum_{i=1}^{wideth(I)-2} \sum_{j=1}^{height(I)-2} GD(i,j)}{(width(I)-2) \times (height(I)-2)}$$

(18)

$$Avg_E(GD_E(i,j)) = \frac{\sum_{i=1}^{wideth(I)-2} \sum_{j=1}^{height(I)-2} GD_E(i,j)}{(width(E)-2) \times (height(E)-2)}$$

(19)

where *I* is the original image and *E* is encrypted image.

Now that Average *GD* is obtained for both original image and scrambled/encrypted image. The value of Gray Difference Degree *GDD* can be determined. Using Equation (20) *GDD* can be determined between original image and scrambled/encrypted image.

$$GDD = \frac{Avg_E(GD_E(i,j)) - Avg(GD(i,j))}{Avg_E(GD_E(i,j)) + Avg(GD(i,j))} \tag{20}$$

*GDD* values were computed for same set of encrypted image used to find correlations, that is $E_{im} = e\left(I'_{im}, L_0^{im}, v_{CGL}, PBC, 12, 6\right)$. *GDD* values in Table 4 demonstrate the technique ability to well scramble images.

4.1.4. Comparisons

In this section a comparison is performed between the proposed XOR-CGL image encryption scheme and other methods on the literature using the differential and statistical metrics. Table 5 shows *NPCR, UACI* and information entropy comparison between the proposed scheme and other methods on literature and Table 6 compares correlation. Comparisons indicate that *NPCR* values were high and *UACI* were close to ideal 33%. Information entropy varied and improvement is on order. As for correlation the scheme produced required week correlation across different tested orientations.

**Table 5.** *NPCR* and *UACI* comparison between proposed scheme and other algorithms.

| Author | Year | NPCR% | | | UACI% | | | Info. Entropy | | |
|---|---|---|---|---|---|---|---|---|---|---|
| | | Lena | Cameraman | Peppers | Lena | Cameraman | Peppers | Lena | Cameraman | Peppers |
| [46] | 2020 | 99.786 | 99.791 | - | 30.325 | 27.637 | - | 7.994 | 7.994 | - |
| [47] | 2021 | 99.62 | 99.63 | - | 33.50 | 33.56 | - | 7.996 | - | - |
| [48] | 2021 | 99.624 | - | 99.603 | 33.422 | - | 33.427 | - | - | 7.997 |
| [49] | 2022 | 99.646 | 99.588 | 99.65 | 33.439 | 33.505 | 33.455 | 7.997 | 7.997 | 7.997 |
| Proposed | 2023 | 99.597 | 99.467 | 99.489 | 28.07 | 37.075 | 31.934 | 6.873 | 6.049 | 7.571 |

**Table 6.** Correlation comparison between proposed scheme and other algorithms.

| Author | Year | Correlation | | | | | | | | |
|---|---|---|---|---|---|---|---|---|---|---|
| | | Lena | | | Cameraman | | | Peppers | | |
| | | Hor. | Ver. | Dia. | Hor. | Ver. | Dia. | Hor. | Ver. | Dia. |
| [48] | 2021 | 0.0069 | - | - | −0.0036 | 0.0048 | 0.0073 | −0.017 | −0.0334 | −0.0073 |
| [49] | 2022 | −0.0035 | 0.0076 | −0.0026 | −0.0252 | −0.006 | −0.0078 | - | - | - |
| [50] | 2021 | - | 0.0479 | 0.0075 | - | - | - | 0.0211 | 0.0129 | 0.0013 |
| [51] | 2021 | 0.0031 | 0.0005 | 0.0003 | 0.002 | 0.0016 | 0.0013 | 0.0015 | 0.0029 | −0.0019 |
| Proposed | 2023 | −0.041 | −0.017 | −0.031 | 0.032 | 0.127 | −0.083 | −0.027 | −0.104 | 0.022 |

*GDD* metric was used to indicate the robustness and effectiveness of various image scrambling techniques in related literature. With that *GDD* metric can be utilized for comparing the proposed technique with other scrambling algorithms given that same test images are used. Table 7 shows a comparison between *GDD* of proposed scheme and other CA based scrambling techniques. Note some of test images were replaced to provide a *GDD* comparison across more images as compared scrambling algorithms were evaluated with different images by their authors. The results show that the proposed algorithm had better *GDD* on most of test images.

**Table 7.** Comparing different CA based scrambling techniques GDDs.

| Author | Method | Lena | Cameraman | Barbara | Gold Hill | Man | 5.1.12 | 7.1.04 |
|--------|--------|------|-----------|---------|-----------|-----|--------|--------|
| [15] | Game of Life | 0.9320 | 0.8971 | **0.8749** | - | - | **-** | - |
| [19] | Elementary CA | 0.9311 | 0.8926 | - | - | - | - | - |
| [24] | Elementary CA | - | 0.8780 | 0.8680 | 0.9000 | 0.8460 | 0.8980 | 0.8810 |
| [52] | Game of Life | 0.9200 | 0.8954 | - | - | **0.9590** | 0.9317 | 0.9456 |
| **Proposed Method** | **XOR-CGL** | **0.9650** | **0.9793** | 0.8117 | **0.9531** | 0.9460 | **0.9234** | **0.9772** |

### 4.2. Encrypted Faces Recognition with LDA Classification

Based on the results of the previous subsection, it is clear that the encryption technique is effective, entire ORL dataset is encrypted with single key $E_{im} = e\left(I'_{im}, L_0, v_{CGL}, PBC, 15, 9\right)$. Figure 9 shows samples of encrypted subjects' images. In recognition pipeline 80% of encrypted subject's images are reserved for training the model. Remaining 20% is used for testing.

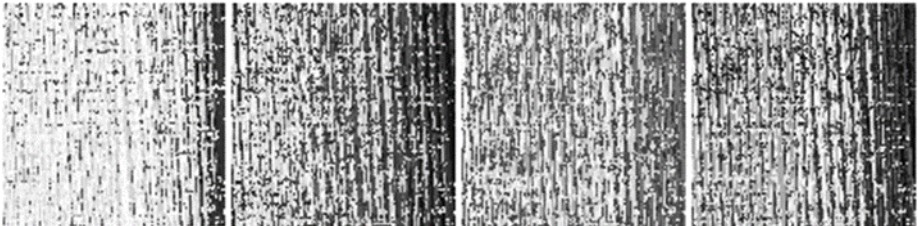

**Figure 9.** Samples of ORL's subjects encrypted face images.

Following encryption of ORL dataset, entire dataset is scaled with standard scaling for elimination of any potential bias in data. Next after splitting of dataset, training set is processed with LDA for extraction of features required for classification of test data. Note that in ORL dataset there are 40 subjects and each subject has 10 samples, 8 of which are used for training and 2 for testing resulting in 320 training sample and 80 testing samples.

Applying random forest classifier, the model was able to achieve 96.25% accuracy. Metrics evaluated are precision, recall, F1-score and accuracy. Those metrics defined by [53] are detailed in Table 8. True positive, true negative, false positive and false negative are given symbols $tp$, $tn$, $fp$ and $fn$ respectively.

**Table 8.** Evaluation metrics in classification report.

| Metric | Formula |
|--------|---------|
| Precision | $precision = \frac{tp}{tp+fp}$ (21) |
| Recall | $recall = \frac{tp}{tp+fn}$ (22) |
| F1-score | $F_1 = \left(1 + \beta^2\right) \times \frac{precision \times recall}{\beta^2 \times precision + recall}$ (23) |
| Accuracy | $Accuracy = \frac{tp+tn}{tp+tn+fp+fn}$ (24) |

Other metrics in Table 9 are support, macro average and weighted average. Since support value is the same for all classes as 2 test sample are provided per class the values

for macro average and weighted average are the same as arithmetic mean and would have no effect in computing corresponding metrics in classification report.

**Table 9.** Evaluation metrics report with random forest classifier. Highlighted in blue are results in encrypted test set and orange represents results on original test set.

| | Precision | | Recall | | F1-Score | | Support | |
|---|---|---|---|---|---|---|---|---|
| | **Enc. Test Set** | **Ori. Test Set** | **Enc. Test Set** | **Ori. Test Set** | **Enc. Test Set** | **Ori. Test Set** | **Enc. Test Set** | **Ori. Test Set** |
| Accuracy | | | | | 0.96 | 0.09 | 80 | 80 |
| Macro Average | 0.97 | 0.02 | 0.96 | 0.09 | 0.96 | 0.04 | 80 | 80 |
| Weighted Average | 0.97 | 0.02 | 0.96 | 0.09 | 0.96 | 0.04 | 80 | 80 |

To prove the proposed scheme capability of withstanding spoofing attacks, the same classification test was run again however this time testing test was replaced with original test images i.e., before encryption. This simulates the situation in which an attacker was able to obtain an authenticated person's face image only and didn't encrypt the face image. In this case classification accuracy of the system was low at 8.75% only. The low classification accuracy indicates that the system is not able to classify unencrypted features and authenticity can only be gained with correctly classified images. Evaluation metrics results for original test image against encrypted features database are highlighted with orange on Table 9.

## 5. Discussion

Using a model trained to recognize encrypted face images with high accuracy, this paper proposes a solution to spoofing's vulnerability in facial image recognition systems. The implementation of such a model requires developing an image encryption algorithm for encrypting face images used for training the recognition model. This image encryption scheme is based on XOR pixel substitutions and CA pixel scrambling.

To evaluate the encryption performance of the image encryption algorithm, it is necessary to analyze the encrypted images encoded with the encryption scheme. Statistical analysis and differential analysis were used in the analysis. Based on the differential analysis with *NPCR* test, the image encryption scheme produced a high percentage of pixels' difference between the original image and the encrypted image. A *NPCR* of 99% or higher was achieved in all test images. *UACI* was used to perform another differential analysis. On test images, the results on *UACI* fluctuated between 33 % and 15% within the ideal range. A statistical analysis of the data was conducted using the following five tests: histogram, correlation, key analysis, information entropy, and *GDD*. XOR operation on pixels' values changed the histogram in encrypted images, so histogram matching could not identify encrypted images. A very weak correlation was observed between the original and encrypted images for the entire image, as well as in the vertical, horizontal, and diagonal directions, due to a very low similarity between the original and encrypted images. Key analysis in done by firstly determining key space for encryption algorithm which is $u \times \left(2^{size(L_0)}\right)$ where $u$ is size of the set of unique paris of $k$ and $n$ values; then testing sensitivity of algorithm by decrypting an encrypted image with slightly different key. This method had excellent key sensitivity, since no visual information could be extracted when decrypting the same image with slightly different keys. For encrypted grayscale images, the information entropy produced different values. Some were extremely close to 8, while others were lower. Based on *GDD* values, the proposed image encryption scheme produced exceptional results exceeding those obtained from other methods described in related literature.

The following observations were made about the robustness and limitations of the proposed image encryption scheme:

- Image encryption algorithms produce very different images when they change the values of pixels in encrypted images. A very weak correlation was observed in all cases, and *NPCR* values exceeded 99% in every case.
- In this method, the key space is very large, and it grows as the size of the image to be encrypted and the number of evolutions selected for configuring the encryption key increases.
- According to the proposed scheme, *GDD* values were exceeding those found in some related literature on image encryption methods based on CAs for the same images.
- A scheme for encrypting images changed the histogram to resist histogram matching attacks, but changing the pixel values with XOR is not sufficiently secure, as an XORed image would have a similar histogram to an encrypted image, hence a more robust scheme to substitute pixels must be incorporated into the algorithm.
- Both *UACI* and information entropy values can be considered acceptable, but either one can be enhanced with a better pixel's substitution scheme.

By analyzing image encryption algorithms, the algorithm is implemented into the face recognition model pipeline by encrypting the face images used to train the LDA-based model. Several experiments were performed on the model using the ORL dataset. The model's accuracy and spoof-resistance were tested in two main experiments. For the first experiment, the entire ORL dataset was encrypted with the same key, then it was split into 80% for training and 20% for testing. In classifying encrypted face images, the system achieved 96.25% accuracy using a random forest classifier. A second experiment used the same encrypted training set but used original images for testing the model. Only 8.75% of the results were accurate in the second experiment. Since both input face images must be encrypted with the same key for a highly accurate recognition rate to be achieved, the LDA-based recognition system is highly resilient to spoofing attacks.

In information systems containing secret or sensitive information, such a system can be used to authenticate users. Authentication can then be obtained once the user adds the required encryption key configurations as well as capturing the user's face. Whenever a system user face identity is revealed by spoofing, the attacker needs to have the correct encryption key configuration otherwise authenticating the system is very difficult. Table 10 shows sensitivity of the model when the testing set was encrypted with slightly different key $E_{im} = e(I'_{im}, L_0, v_{CGL}, PBC, 14, 9)$ from the one used in encrypted testing set in Table 9, i.e., first experiment. Here the accuracy decreased significantly to 66.25%.

**Table 10.** Evaluation metrics report with random forest classifier for testing set encrypted with slightly different key.

|  | Precision | Recall | F1-score | Support |
|---|---|---|---|---|
| Accuracy |  |  | 0.66 | 80 |
| Macro Average | 0.67 | 0.66 | 0.62 | 80 |
| Weighted Average | 0.67 | 0.66 | 0.62 | 80 |

As in the case of the image encryption scheme, the following points concerning the effectiveness and limitations of LDA based encrypted faces recognition model were observed:

- The proposed LDA based encrypted faces recognition model produced high accuracy in classification of encrypted faces images with the same encryption key reaching an accuracy of 96.25%.
- The model is highly sensitive to encrypted face images with slightly different key. The model accuracy dropped to 66.25% when it was tested with testing set encrypted with a slightly different key.
- The model is able to effectively resist spoofing attacks. Testing model with original images testing set showed that the model achieved 8.75% accuracy only.

- The model security is limited with robustness of image encryption scheme used. The weakness of the image encryption scheme introduces vulnerabilities to encrypted faces recognition model.
- The image encryption scheme needs to be robust enough to provide effective encryption performance however; the image encryption scheme must retain enough features in resulting encrypted images in order for the model to distinguish between different classes.

## 6. Conclusions

In conclusion, biometric based recognition systems including face recognition are vulnerable to spoofing attacks in which an attacker could assume the identity of authenticated individual by obtaining an artificial copy of that individual's biometric. The solution proposed in this paper is integration of image encryption scheme into face recognition pipeline. This addition in recognition pipeline requires the attacker to submit an encrypted copy of same individual face image with correct key used for encryption of features database in order to gain false authentication.

Experimentation was performed for XOR-OTCA CGL image encryption scheme firstly to prove its robustness. Differential and statistical analysis showed that all testing images had more than 99% *NPCR*, correlation at almost 0, and high values for *GDD* metric. Key space was $u \times \left( 2^{res(L_0)} \right)$ and key is sensitive to slight changes as no useful information can be extracted from images decrypted with slightly different keys. After proving the robustness of image encryption scheme, experimentations were performed on LDA based face recognition scheme with integrated image encryption scheme. Testing showed that the proposed pipeline had an accuracy of 96.25% in classifying encrypted test face images from encrypted features database on ORL dataset. The same test was conducted with ORL original test face images against encrypted features database; this time the accuracy was low at 8.75% which proves the proposed scheme capability to withstand spoofing attacks.

**Author Contributions:** Conceptualization, E.A. and B.I.; Methodology, E.A. and B.I.; Software, B.I.; Validation, K.Z.; Formal analysis, E.A. and B.I.; Investigation, E.A. and B.I.; Resources, E.A., B.I. and K.Z.; Data curation, E.A. and B.I.; Writing—original draft, E.A. and B.I.; Writing—review & editing, E.A., B.I., K.Z. and M.R.; Visualization, B.I., K.Z. and M.R.; Supervision, E.A.; Project administration, E.A. All authors have read and agreed to the published version of the manuscript.

**Funding:** This research received no external funding.

**Data Availability Statement:** The data presented in this study are publicly available through the University of Waterloo's image repository [Repository (uwaterloo.ca)], and University of South California SIPI Institute miscellaneous volume, [SIPI Image Database—Misc (usc.edu)]. ORL dataset was obtained from Kaggle [AT&T Database of Faces | Kaggle].

**Conflicts of Interest:** The authors declare no conflict of interest.

## Appendix A

**Table A1.** Recorded *NPCR* and *UACI* values on Yale dataset.

| | Subject 1 | | | | | | | | | | |
|---|---|---|---|---|---|---|---|---|---|---|---|
| NPCR% | 99.723 | 99.679 | 99.702 | 99.298 | 99.708 | 99.694 | 99.561 | 99.689 | 99.734 | 99.727 | 99.7 |
| UACI% | 38.129 | 37.682 | 38.222 | 30.642 | 38.454 | 38.073 | 21.891 | 38.198 | 37.928 | 38.266 | 37.867 |
| | Subject 2 | | | | | | | | | | |
| NPCR% | 99.734 | 99.762 | 99.696 | 99.623 | 99.727 | 99.727 | 99.668 | 99.752 | 99.716 | 99.739 | 99.752 |
| UACI% | 40.962 | 38.875 | 38.902 | 35.887 | 39.501 | 39.501 | 33.462 | 39.081 | 39.493 | 38.403 | 39.184 |

**Table A1.** *Cont.*

| | | | | | | | | | | | |
|---|---|---|---|---|---|---|---|---|---|---|---|
| | | | | | | Subject 3 | | | | | |
| NPCR% | 99.763 | 99.769 | 99.792 | 99.344 | 99.817 | 99.817 | 99.438 | 99.79 | 99.781 | 99.808 | 99.784 |
| UACI% | 42.674 | 44.391 | 43.901 | 36.389 | 43.613 | 43.613 | 26.951 | 43.55 | 43.628 | 43.403 | 43.495 |
| | | | | | | Subject 4 | | | | | |
| NPCR% | 99.737 | 99.779 | 99.765 | 99.603 | 99.74 | 99.747 | 99.361 | 99.747 | 99.785 | 99.75 | 99.764 |
| UACI% | 41.591 | 40.895 | 41.163 | 34.745 | 41.251 | 41.097 | 24.89 | 41.097 | 40.955 | 40.947 | 41.289 |
| | | | | | | Subject 5 | | | | | |
| NPCR% | 99.65 | 99.738 | 99.739 | 99.542 | 99.771 | 99.76 | 99.552 | 99.767 | 99.747 | 99.801 | 99.796 |
| UACI% | 36.549 | 38.378 | 38.485 | 31.218 | 38.229 | 38.311 | 22.701 | 38.329 | 38.181 | 38.69 | 38.43 |
| | | | | | | Subject 6 | | | | | |
| NPCR% | 99.693 | 99.776 | 99.709 | 99.608 | 99.754 | 99.754 | 99.767 | 99.722 | 99.776 | 99.758 | 99.759 |
| UACI% | 41.645 | 39.221 | 38.18 | 32.146 | 38.26 | 38.26 | 43.915 | 38.666 | 38.855 | 39.134 | 38.865 |
| | | | | | | Subject 7 | | | | | |
| NPCR% | 99.748 | 99.776 | 99.769 | 99.235 | 99.777 | 99.777 | 99.439 | 99.781 | 99.755 | 99.756 | 99.8 |
| UACI% | 42.273 | 44.19 | 44.254 | 35.767 | 43.831 | 43.831 | 33.181 | 44.636 | 44.004 | 44.208 | 44.247 |
| | | | | | | Subject 8 | | | | | |
| NPCR% | 99.757 | 99.832 | 99.82 | 99.405 | 99.793 | 99.832 | 99.619 | 99.808 | 99.811 | 99.813 | 99.794 |
| UACI% | 42.001 | 43.325 | 41.585 | 33.345 | 42.078 | 43.325 | 34.071 | 40.547 | 41.728 | 41.708 | 41.193 |
| | | | | | | Subject 9 | | | | | |
| NPCR% | 99.742 | 99.803 | 99.769 | 99.595 | 99.768 | 99.768 | 99.641 | 99.805 | 99.785 | 99.808 | 99.803 |
| UACI% | 41.498 | 41.478 | 41.776 | 34.993 | 41.728 | 41.728 | 35.227 | 42.406 | 41.953 | 41.877 | 41.737 |
| | | | | | | Subject 10 | | | | | |
| NPCR% | 99.729 | 9.787 | 99.804 | 99.553 | 99.76 | 99.779 | 99.426 | 99.765 | 9.764 | 99.757 | 99.747 |
| UACI% | 42.121 | 43.653 | 43.969 | 34.986 | 43.578 | 43.731 | 39.447 | 43.724 | 43.844 | 42.946 | 43.403 |
| | | | | | | Subject 11 | | | | | |
| NPCR% | 99.634 | 99.706 | 99.732 | 99.733 | 99.757 | 99.742 | 99.421 | 99.657 | 99.731 | 99.706 | 99.706 |
| UACI% | 42.714 | 42.278 | 42.765 | 37.832 | 42.808 | 42.657 | 39.969 | 41.079 | 42.562 | 42.76 | 42.389 |
| | | | | | | Subject 12 | | | | | |
| NPCR% | 99.687 | 99.718 | 99.645 | 99.615 | 99.729 | 99.729 | 99.74 | 99.667 | 99.652 | 99.686 | 99.655 |
| UACI% | 37.653 | 37.55 | 36.404 | 34.901 | 38.363 | 38.363 | 40.079 | 36.028 | 35.311 | 36.109 | 36.036 |
| | | | | | | Subject 13 | | | | | |
| NPCR% | 99.837 | 99.824 | 99.819 | 99.684 | 99.837 | 99.801 | 99.564 | 99.829 | 99.84 | 99.85 | 99.841 |
| UACI% | 45.053 | 44.111 | 44.262 | 38.299 | 44.316 | 44.213 | 34.996 | 44.252 | 44.297 | 43.954 | 44.47 |
| | | | | | | Subject 14 | | | | | |
| NPCR% | 99.719 | 99.81 | 99.779 | 99.702 | 99.779 | 99.779 | 99.721 | 99.784 | 99.794 | 99.777 | 99.8 |
| UACI% | 42.373 | 43.286 | 43.574 | 33.404 | 43.198 | 43.198 | 36.772 | 43.779 | 43.192 | 43.556 | 43.274 |
| | | | | | | Subject 15 | | | | | |
| NPCR% | 99.761 | 99.745 | 99.793 | 99.218 | 99.8 | 99.777 | 99.6 | 99.789 | 99.779 | 99.774 | 99.784 |
| UACI% | 41.005 | 41.252 | 43.096 | 33.5 | 42.052 | 43.066 | 32.021 | 42.633 | 41.855 | 42.053 | 42.644 |

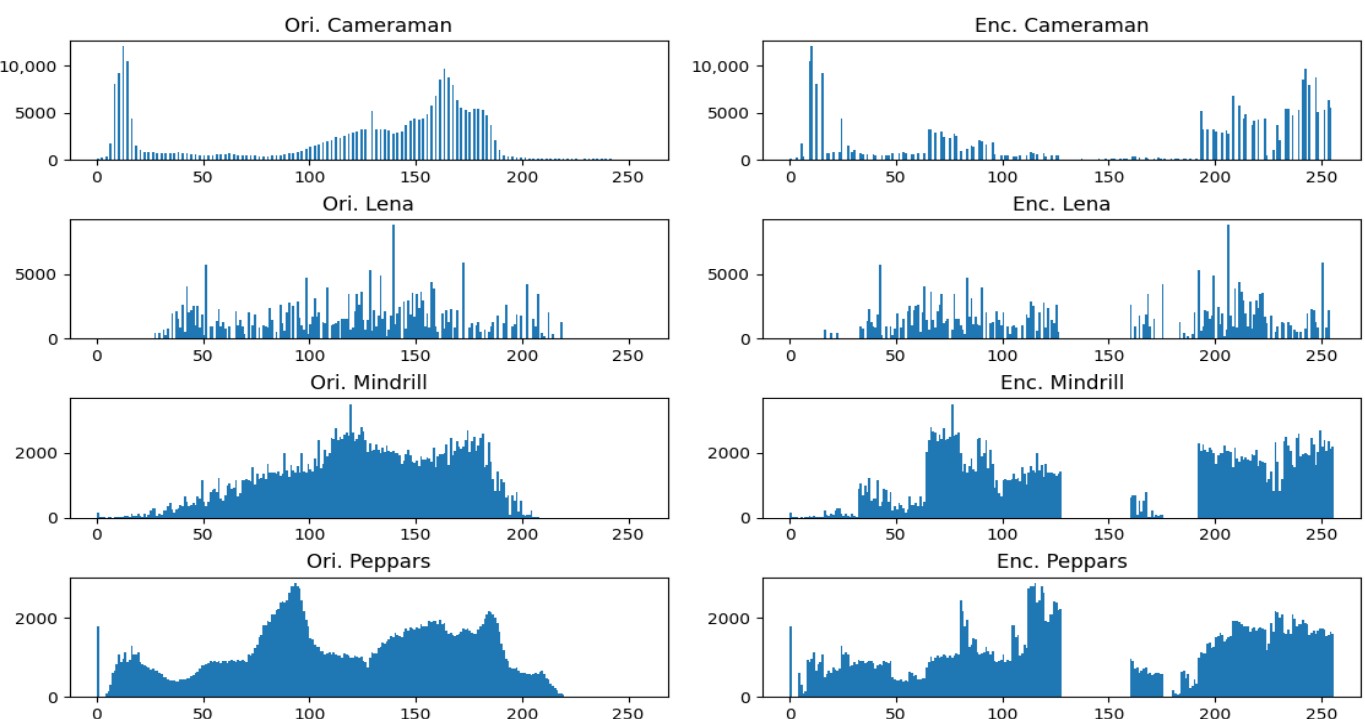

**Figure A1.** Histograms of multiple test images. On the left are original images histograms and on the right are encrypted images histograms.

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
