# Peer review of "Facial Image Encryption for Secure Face Recognition System"

_electronics, doi:10.3390/electronics12030774_

Round 1

Reviewer 1 Report

This paper deals with facial image encryption for a secure face recognition system in which an image encryption scheme to counter spoofing attacks by integrating it into the pipeline of Linear Discriminant Analysis (LDA) based face recognition is proposed. my observations for this work are as follows:

1. Literature survey is not up to the mark, as various methods have been developed for encryption in image recognition. What is the need analysis?

2. Robustness of the proposed scheme is also unclear.

3. Please provide a comparative study to justify the viability of the proposed method.

4. Fig. 1 is well-known and can be removed.

Reviewer 2 Report

The present paper proposes an image encryption scheme to counter spoofing attacks by integrating it into a pipeline using
Linear Discriminant Analysis for face recognition. The method is based on XOR pixels substitution and cellular automata for scrambling.                                                                               
The paper is well written in English and it presents an informative state of the art, which is pertinent to introduce the problem

My main concerns with the present research are the following:

a) Please insert a caption in all the presented Algorithms.
b) Please eliminate the "#" in equations, please refers according (x).
c) Please insert the number in equations. Different sections such as 3.1.2.1.
d) In Figure 2, Improve "Forset" by "Forest"
e) In Table 1, please present NPCR in a big database, not only the presented well-known images.
f) Please improve the quality of Figure 5.
g) Table 1,2,3 can be organized in just one table.
h) In Table 4, the column "Definition" is not required.
i) Table 5 and Table 6, can be organized in just one table and only present statistical analysis not the 39 subjects.
j) Please present illustrative images of the experiments
k) Please present a comparative analysis between the proposed method and methods of the state of the art.
l) Please justify the use of LDA, instead of another machine learning techniques.

Reviewer 3 Report

This study is to explore facial image encryption for secure face recognition systems using a linear discriminant analysis. The topic is timely and the study purpose is clear. Overall study processes are valid and proper. The Following are some comments to improve the quality of the paper.

- Simplify figure 1. There are some redundant words and figures.  

- The study says that there are 6 stages in face recognition. But, fig. 1 shows 7 stages. Make a correction accordingly.

- In figures 3 and 4, scrambled images are not meaningful to readers. If there is any way to be meaningful, please do so.  

- If possible, add some statistical values to justify the study results.

- Before, 5. Conclusion section, add more  implication of this study.

- Double check the manuscript to be professional.

Reviewer 4 Report

1.     Research Problem, main goal, motivation, and contributions should be highlighted in introduction Section.  

2.     In Figure 1, the authors mentioned the feature DB, however, it is not clear what is DB.

3.     Referencing should be done correctly, check paragraph 3 in Introduction Section.

4.     Highlight and formulate the research problem in the introduction Section.

5.     Contributions in the introduction section should be highlighted (write contributions in a point structure).

6.     The third paragraph in the related work section is missing referencing.  

7.     Check Equations (1 and 2), what is the difference between them?

8.     Revise equation 4 and denote all parameters in equation, also explain why (#1).

9.     All algorithm should be named in the manuscript.

10.  What is the difference between (X) in equation (3 and 5), also same thing for b in equations (1, 2, and 6)?

11.  Note that all equations must be numbered in manuscript.

12.  (I See #1, #2, #3, to #9) but it is not clear why?

13.  It is not clear how the secret key has been generated in this work.

14.  Significant of this work should be highlighted.

15.  More analysis should be provided such as (the value key space should be presented), (histogram of more images), (correlation as horizontal, vertical, and diagonal), (information entropy), and (UACI).

16.  In subsection 4.1.3.3. there are some steps how to compute GDD, it is better to set the Equation of GDD and all parameters should be explained, no need to write an algorithm for computing GDD.

17.  A discussion section should be added.  

18.  Limitations of the proposed method should be highlighted.

19.  The paper is not structured well, the authors should revise their manuscript, and explain why encryption and classification together its confusing.  

20.  Compare your results with most recent previous works. 

Round 2

Reviewer 1 Report

The responses are addressed. Thank You.

Author Response

Thank you for your review.

Reviewer 2 Report

A comparison between the proposed method and methods of the state-of-the-art is required. This comparison should be performed in all the presented images (Table 5 is not enough ).

Reviewer 4 Report

1.     Check and revise the first and second of your contributions.

2.     Give names to Algorithms on the top of algorithms with their captions.

3.      The generated key is still not clear, authors should justify the robustness of the key.

4.     The NIST test is required to show the randomness of the proposed method.

5.     Comparison is required with the most recently published works.

Round 3

Reviewer 2 Report

My comments have been properly addressed.

Author Response

Thank you for your review

Reviewer 4 Report

Please make sure your paper should prepare with the MDPI template.
